

# Discriminating three motor imagery states of the same joint for brain-computer interface

Shan Guan, Jixian Li, Fuwang Wang, Zhen Yuan, Xiaogang Kang and Bin Lu

School of Mechanical Engineering, Northeast Electric Power University, Jilin, China

## ABSTRACT

The classification of electroencephalography (EEG) induced by the same joint is one of the major challenges for brain-computer interface (BCI) systems. In this paper, we propose a new framework, which includes two parts, feature extraction and classification. Based on local mean decomposition (LMD), cloud model, and common spatial pattern (CSP), a feature extraction method called LMD-CSP is proposed to extract distinguishable features. In order to improve the classification results multi-objective grey wolf optimization twin support vector machine (MOGWO-TWSVM) is applied to discriminate the extracted features. We evaluated the performance of the proposed framework on our laboratory data sets with three motor imagery (MI) tasks of the same joint (shoulder abduction, extension, and flexion), and the average classification accuracy was 91.27%. Further comparison with several widely used methods showed that the proposed method had better performance in feature extraction and pattern classification. Overall, this study can be used for developing high-performance BCI systems, enabling individuals to control external devices intuitively and naturally.

## INTRODCTION

BCI is a technology that can directly establish communication and control between human brain and computer or other electronic equipment (*Li et al., 2020*; *Park & Chung, 2020*). BCI technology is widely applied in medical rehabilitation, smart home, entertainment, military, and other fields. At present, the EEG traces include sensorimotor rhythms (SMR) (*Blankertz et al., 2010*), slow cortical potential (SCP) (*Hinterberger et al., 2004*), event-related potential (ERP) (*Delgado et al., 2020*), and visual-evoked potential (VEP) (*Zhou et al., 2020*). SMR is induced by motor imagery without external stimulation, so it is widely used in BCI systems. However, the EEG is a non-stationary, non-linear, and noisy signal. And, it is easily interfered by the environment when it is recorded, the signal-to-noise ratio is low, which makes it more challenging to identify the EEG of MI tasks.

Corresponding author
Jixian Li, 2201900511@neepu.edu.cn, lijixian1107@163.com

In the past few decades, researchers have proposed various feature extraction methods and classification algorithms to classify MI tasks efficiently. The most classical feature extraction methods include wavelet transform (WT) (*You, Chen & Zhang, 2020*), empirical mode decomposition (EMD) (*Taran et al., 2018*), common spatial pattern (CSP) (*Yang et al., 2016*; *Selim et al., 2018*), and filter-bank CSP (FBCSP) (*Ang et al., 2008*; *Wang et al., 2020*). The widely used classification algorithms include linear discriminant analysis (LDA) (*Aljalal, Djemal & Ibrahim, 2019*), extreme learning machine (ELM) (*Rodriguez-Bermudez, Bueno-Crespo & Martinez-Albaladejo, 2017*), k-nearest neighbors (KNN) (*Bashar, Hassan & Bhuiyan, 2015*), support vector machine (SVM) (*Selim et al., 2018*) and least squares support vector machine (LS-SVM) (*Taran et al., 2018*; *Taran & Bajaj, 2019*). Malan and Sharma applied dual-tree complex wavelet transform (DTCWT) to extract time, frequency, and phase features of left and right hand MI EEG signals, and classified them using SVM with an average accuracy of 80.7% (*Malan & Sharma, 2019*). However, it is usually hard to select an appropriate wavelet basis function. *Taran et al. (2018)* employed EMD to extract MI features of left and right hands, and classified them by using LS-SVM with an average accuracy of 97.56%. The weakness of that study is that it ignored the endpoint effect and mode mixing phenomenon in the EMD process. *Miao, Wang & Liu (2017)* classified left and right hand movements with an average accuracy of 86.41%, by using sparse representation of CSP features. *Kumar & Sharma (2018)* proposed a parameter tuning algorithm to improve the performance of CSP by selecting the optimal filter parameters. That study reported an average error recognition rate of 10.19% on BCI Competition III Dataset IVa (right hand and foot). Recently, *Kumar, Sharma & Sharma (2021)* used genetic algorithm (GA) for adaptive filtering, combined CSP and long short-term memory network (LSTM) for feature extraction, and applied SVM for classification. It should be noted that these studies focused on binary problems. The aforementioned methods have achieved good recognition results for the classification of the left hand, right hand, foot, and tongue MI tasks. When imagining movements of different limbs, event-related de-synchronization/event-related synchronization (ERD/ERS) will occur in the corresponding areas of the motor cortex. This phenomenon can be recorded by the different electrodes overlying the motor cortex, so BCI systems can efficiently identify MI tasks within different limbs.

However, ERD/ERS induced by MI tasks of the same limb usually occurs in the adjacent region of the motor cortex, which makes it exceedingly difficult to detect the MI tasks within the same joint. Recently, a small number of researchers have conducted preliminary explorations of the recognition of MI tasks within the same joint. For instance, *Vuckovic & Sepulveda (2012)* employed Gabor coefficients calculated from independent components as features and Elman's neural networks as the classifier, the average recognition accuracy of wrist extension and flexion MI was 67.5%. As we all know, that accuracy cannot meet the working requirements of BCI systems. *Edelman, Baxter & He (2016)* used EEG source imaging technology to identify 4-class MI tasks (wrist flexion, extension, pronation, and supination) with the recognition accuracy of each class exceeding 79.00%.

From the above analysis, we can see that most existing studies focus on the MI tasks of different limbs, which have achieved good recognition results. However, these methods
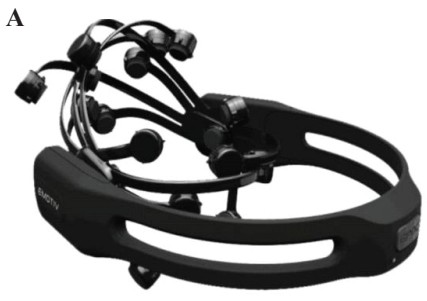
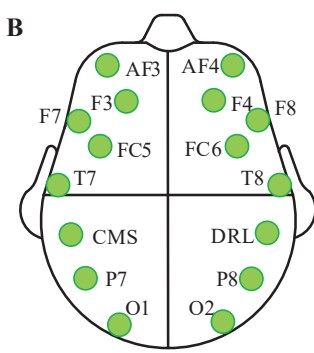

**Figure 1** **Emotive Epoc+ and the placement position of Emotive electrodes.** (A) Emotive Epoc+ EEG signal acquisition instrument. (B) Location of all electrodes and marked electrodes are used in this study.

can be further improved in some aspects. Furthermore, fewer studies decode multiclass MI tasks of the same joint, and the recognition accuracy of the same joint is lower than that of different limbs. Therefore, this paper proposes a method to decode the various MI movements of the same joint using LMD-CSP and MOGWO-TWSVM. First, LMD is used to decompose the preprocessed MI EEG into a series of product functions (PFs). According to the entropy (En) and super entropy (He) of the cloud model, the real PF components are selected. Second, the selected PFs of each channel are reconstructed into a new signal matrix, then feature vectors are extracted by using the CSP. Finally, TWSVM optimized by the MOGWO algorithm is applied to discriminate the extracted features. The proposed framework was verified using our laboratory data sets, which include shoulder abduction, extension, and flexion MI tasks. There were two reasons why we chose these movements. First, these movements were part of the exercise for stroke patients. Second, these movements could intuitively control the robotic arm. In addition, we compared the LMD-CSP and MOGWO-TWSVM with several widely used feature extraction methods and classifiers, respectively. The conventional feature extraction methods include time domain parameters (TDP) (*Tavakolan et al., 2017*), CSP (*Selim et al., 2018*), FBCSP (*Ang et al., 2008*), and common spatial pattern based on empirical mode decomposition (EMD-CSP) (*Wang et al., 2008*). The traditional classifiers include LDA, ELM, KNN, SVM, and LS-SVM. Those comparisons could confirm the quality of the proposed method.

## MATERIALS & METHODS

### EEG signal acquisition

Emotive Epoc+ was employed to record the EEG signals. It includes 14 effective electrodes (AF3, F7, F3, FC5, T7, P7, O1, O2, P8, T8, FC6, F4, F8, and AF4) and 2 reference electrodes (CMS, DRL). The sampling rate is 128 Hz. The layout of electrodes is according to the standard international 10–20 system. The equipment and electrodes arrangement are shown in Fig. 1.

Seven right-handed subjects (five males and two females aged from 23 to 28), who were in good condition both psychologically and physically, were randomly selected to
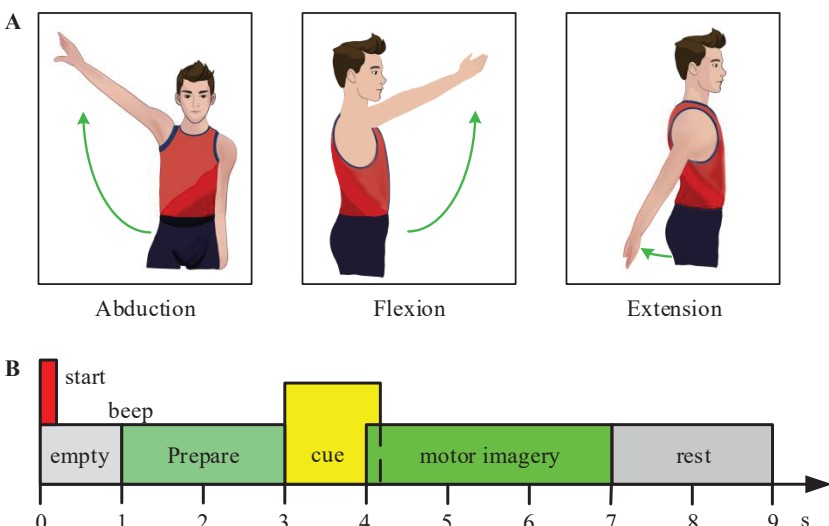

**Figure 2  Experimental paradigm for the motor imagery (MI) tasks.** (A) The experiment included three MI tasks (shoulder abduction, flexion, and extension). (B) The experimental process of one trail.

participate in this experiment. All subjects had given written informed consent before the experiment. The study was approved by the Scientific Research Ethics and Technology Safety committee of Northeast Electric Power University.

In the experiment, the subjects were required to perform 3-class MI tasks (shoulder abduction, extension, and flexion) according to prompts. Figure 2 shows the experimental paradigm for the motor imagery (MI) tasks. At the start of each trial, the subjects naturally placed their hands and kept relaxed. At time point t =1s, the monitor showed a cross '+', and presented a short beep tone to raise the subject's attention. At t =3s, an arrow pointed to the left, right, or top at random, was shown to indicate the subjects to imagine the corresponding movement (shoulder abduction, extension, or flexion movement). And the arrow disappeared after 1.25 s. At t =7s, the subjects stopped motor imagination. The next trial began after 2 s. For training classifiers, 60 trials per subject were collected in total (20 trials per class).

## EEG preprocessing

First, the raw EEG data were filtered between 8 Hz and 30 Hz by a 5th order zero-phase Butterworth filter to remove DC drift and high-frequency noise. Second, the automatic artifact removal (AAR) toolbox was used to remove electrooculogram (EOG) and electromyogram (EMG) artifacts. Third, the common average reference (CAR) was used to reduce the background noise. The CAR method is the selected channel minus the average of all electrodes. Finally, the denoised EEG data were processed by mirror extending technology to eliminate the influence of the endpoint effect in the LMD process. The extended sequence is defined as follows:

$$x_i(t) = [s_i(n), s_i(n-1), \ldots, s_i(1), s_i(1), \ldots, s_i(n), s_i(n), \ldots, s_i(1)]^{\mathrm{T}} \tag{1}$$

where $s_i(c)$ represents the denoised EEG signal of the $i$th electrode.

## Research on feature extraction method based on LMD-CSP
### Local mean decomposition
LMD is a new adaptive time-frequency analytical algorithm first proposed by *Smith (2005)*. When compared with EMD, LMD can better reduce the mode mixing phenomenon and has lower computational complexity and higher decomposition speed (*Zhang & Chen, 2017*). Using the LMD method, the extended sequence $x_i(t)$ is decomposed into a set of PFs and a residual component, and the expression is as follows:

$$x_i(t) = \sum_{j=1}^{k} h_{ij}(t) + u_i(t) \tag{2}$$

where $h_{ij}(t)$ is the $j$th PF obtained from the $i$th electrode, and $u_i(t)$ represents the residual component.

### Selection of product functions based on the cloud model
The cloud model is an uncertainty transformation model based on fuzzy set theory and probability theory, which can achieve the transformation between a qualitative concept and its quantitative data. It expresses the qualitative concept through the three digital characteristics {Ex, En, He}. Expectation (Ex) is the central value of a concept, entropy (En) represents the randomness of a qualitative concept, and super entropy (He) is the dispersion degree of a concept (*Wang et al., 2019*).

$$Ex = \frac{1}{n} \sum_{i=1}^{n} x_i \tag{3}$$

$$En = \sqrt{\frac{\pi}{2}} \times \frac{1}{n} \sum_{i=1}^{n} |x_i - Ex| \tag{4}$$

$$He = \sqrt{S - En^2} \tag{5}$$

where $S$ is the second-order central moment of $x_i$, $x_i(i = 1, 2, \ldots, n)$ represents the quantitative value of n cloud-droplets.

En and He can express the degree of complexity. The larger the values of En and He, the more complex the signals, and vice versa. Since the structure of the real EEG components is generally complicated, the En and He of the real PFs are larger than those of false PFs. Therefore, the parameters En and He of the cloud model are used to select the effective PFs. In our study, through analyzing a large amount of experimental data, we selected the PF1 component as the effective component for subsequent processing. The details of the effective PF selection can be found in the Supplemental Information. For the recognition of other data sets, we need to employ the entropy (En) and super entropy (He) of the cloud model to select effective PF components again.

In this research, the experimental data of all channels were decomposed by LMD in turn, and the PF1 components of each channel were recombined to construct a new signal matrix $X \in R^{M \times N}$, where M is the order of the selected PFs, and N represents the sampling point of PF. In the next step, CSP was used to extract the spatial features of $X$.

### Common spatial pattern

The goal of CSP is to design a spatial filter which can maximize the variance of two kinds of motor imagery EEG data. Because CSP is based on the simultaneous diagonalization of a 2-class covariance matrix, it can only be used in binary problems. Therefore, we applied the one-versus-one (OVO) scheme to LMD-CSP for the multiclass problem, so that a $k$-class problem was transformed into $k$ ($k$-1)/2 binary class problems. For the signal matrix $X_i$ of the $i$ th experimental data, the covariance matrix is calculated as follows:

$$R = \frac{X_i X_i^{\mathrm{T}}}{\mathrm{trace}(X_i X_i^{\mathrm{T}})} \tag{6}$$

where "T" represents the transpose operator, and trace($\cdot$) means to find the trace of a matrix.

For the binary class MI tasks (*i.e.,* shoulder abduction and extension), we calculated the covariance matrix over the trails of each class, and averaged them to obtain the mean covariance matrix $R_A$ and $R_E$. Then the $R_A$ and $R_E$ were transformed to obtain the spatial filter $W_{AE}$ (in this paper, $W_{AE} \in \mathrm{R}^{8 \times 14}$) (*Liu et al., 2012*). Combined with the OVO scheme, we obtained three spatial filters ($W_{AE}$, $W_{AF}$ and $W_{EF}$). Then the three spatial filters were spliced vertically to gain a global spatial filter. The global spatial filter is as follows:

$$W = [W_{AE}; W_{AF}; W_{EF}] \tag{7}$$

where $W_{AE}$ represents the spatial filter between abduction and extension, $W_{EF}$ represents the spatial filter between extension and flexion, and $W_{AF}$ represents the spatial filter between abduction and flexion.

For a single trial data, we obtained the LMD-CSP projection matrix $Z = WX_i$. The $p$ th row of $Z$ is denoted as $Z_p$. The required characteristics can be obtained by:

$$f_p = \log\left(\frac{\mathrm{var}(Z_p)}{\sum_{k=1}^{g} \mathrm{var}(Z_k)}\right) \tag{8}$$

where $p = 1, 2, \ldots g$. The features obtained by LMD-CSP are denoted as $F = [f_1, f_2, \ldots f_g]$.

Figure 3 shows the distributions of the most significant two LMD-CSP features from subjects 1. Obviously, the features of different MI tasks extracted by LMD-CSP are highly distinguishable and these are easy to separate.

## Research on classification method based on MOGWO-TWSVM
### Twin support vector machine

TWSVM, first proposed by *Jayadeva, & Khemchandani & Chandra (2007)* is a new machine learning method based on traditional SVM. TWSVM aims to construct a hyperplane for each class. It requires that each hyperplane is close to the corresponding class samples as possible, and far away from the other class samples as possible. TWSVM has higher training speed and generalization ability than SVM because the former solves two small quadratic programming problems (QPPs) to construct the hyperplanes like SVM, and the constraint condition of a QPP is only related to one class of samples.

Suppose that training samples of class 1 are denoted as $A = [x_1^{(1)}, x_2^{(1)}, \ldots, x_{m1}^{(1)}]^{\mathrm{T}} \in R^{m1 \times g}$, where $x_j^{(i)} \in R^g$ represents the $j$th sample of class 1, $m1$ represents the number of
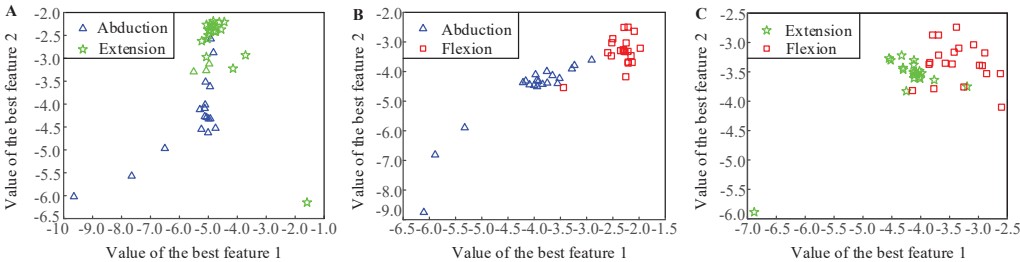

**Figure 3  Distributions of the best two features obtained by the proposed feature extraction method (LMD-CSP).** (A) Distribution of the best two features of abduction and extension. (B) Distribution of the best two features of abduction and flexion. (C) Distribution of the best two features of extension and flexion.

samples. Similarly, training samples of class 2 are denoted as $B \in R^{m2 \times g}$. When training the TWSVM classifier between class 1 and class 2, two nonparallel hyperplanes are obtained as follows:

$$K\left(x^T, C^T\right) w_1 + b_1 = 0 \tag{9}$$

$$K\left(x^T, C^T\right) w_2 + b_2 = 0 \tag{10}$$

where $C^T = [A^T B^T]^T$, $w_1$ and $w_2$ are two normal vectors of hyperplanes, $b_1$ and $b_2$ are the bias vectors.

TWSVM constructs the two hyperplanes by solving the following optimization problems:

$$\min_{w^{(1)}, b^{(1)}, \xi^{(2)}} \frac{1}{2} \left\| K\left(A, C^T\right) w^{(1)} + e_1 b^{(1)} \right\|^2 + c_1 e_2^T \xi^{(2)} \tag{11}$$

$$-\left(K\left(B, C^T\right) w^{(1)} + e_2 b^{(1)}\right) \geq e_2 - \xi^{(2)}, \xi^{(2)} \geq 0$$

$$\min_{w^{(2)}, b^{(2)}, \xi^{(1)}} \frac{1}{2} \left\| K\left(B, C^T\right) w^{(2)} + e_2 b^{(2)} \right\|^2 + c_2 e_1^T \xi^{(1)} \tag{12}$$

$$-\left(K\left(A, C^T\right) w^{(2)} + e_1 b^{(2)}\right) \geq e_1 - \xi^{(1)}, \xi^{(1)} \geq 0$$

where $c_1$ and $c_2$ are penalty parameters, $e_1$ and $e_2$ are column vectors of ones.

In this study, we established a classifier for each binary class, so $k$ ($k$-1)/2 TWSVM classifiers were constructed. In the process of classification, the feature vectors of each MI task were input into the classifiers, and the final result was obtained by voting. The penalty parameters $c_1$ and $c_2$ of the TWSVM were set by the following MOGWO.

### Multi-objective Grey Wolf Optimizer

MOGWO, proposed by *Mirjalili et al. (2016)* is a new swarm intelligence optimization algorithm based on the conventional grey wolf optimizer algorithm. When compared with other multi-objective optimization algorithms, MOGWO has higher convergence and coverage (*Dilip et al., 2018*). In MOGWO, the location of each gray wolf is denoted as a solution, and the first three best solutions are denoted as the alpha ($\alpha$) wolves, beta ($\beta$)

wolves, and delta ($\delta$) wolves, the other candidate solutions are omega ($\omega$) wolves. In the iterative process, $\omega$ wolves are led by $\alpha$ wolves, $\beta$ wolves and $\delta$ wolves to find global optimal solutions.

MOGWO uses an external archive to store and update the non-dominated Pareto optimal solutions. At the same time, it also employs a leader selection mechanism to search for the least crowded segments from the archive, and three non-dominated solutions of the segments are used as $\alpha$, $\beta$, $\delta$ wolves by a roulette-wheel method. The location of each search agent is updated as follows:

$$\vec{D}_i = |\vec{C}_i \cdot \vec{X}_i(t) - \vec{X}(t)|, i \in (\alpha, \beta, \delta) \tag{13}$$

$$\vec{X}_i(t) = \vec{X}_i(t) - \vec{A}_i \cdot \vec{D}_i, i \in (\alpha, \beta, \delta) \tag{14}$$

$$\vec{X}(t+1) = \frac{\vec{X}_\alpha(t) + \vec{X}_\beta(t) + \vec{X}_\delta(t)}{3} \tag{15}$$

where the coefficient vector $\vec{A}$ and $\vec{C}$ are calculated as follows:

$$\vec{A} = 2\vec{a} \times \vec{r}_1 - \vec{a} \tag{16}$$

$$\vec{C} = 2 \cdot \vec{r}_2 \tag{17}$$

where the parameter $\vec{a}$ is decreased from 2 to 0 in the iterative process, and $\vec{r}_1$, $\vec{r}_2$ are random vectors in [0, 1].

### Multi-objective function

The average recognition accuracy is usually employed to measure classification performance. However, it only quantifies the overall classification performance and ignores evaluating the classification results of each class. Therefore, we applied the mean recognition accuracy and the recognition of each class as objective functions to evaluate the candidate solutions generated by the MOGWO algorithm. The objective function can be expressed as:

$$Accuracy = \frac{\sum_{i=1}^3 NC_i}{\sum_{i=1}^3 NC_i + \sum_{i=1}^3 NE_i} \tag{18}$$

$$CR_i = \frac{NC_i}{NC_i + NE_i}, i = 1, 2, 3 \tag{19}$$

where $Accuracy$ is the mean recognition accuracy, $CR_i$ represents the recognition accuracy of $i$ th class, $NC_i$ is the number of $i$ th class correctly distinguished samples, $NE_i$ is the number of $i$ th class wrongly distinguished samples.

**Table 1  Parameter setting used in multi-objective grey wolf optimization (MOGWO).**

| Parameter | Name | Value |
|---|---|---|
| $n$ | Number of wolves | 12 |
| $M_I$ | Max iterations | 100 |
| $A$ | Archive size | 10 |
| $\alpha$ | Grid inflation parameter | 0.1 |
| $\beta$ | Leader selection parameter | 4 |
| $\delta$ | Number of grids per dimension | 10 |

## Statistical analysis

In this study, the one-way analysis of variance (ANOVA) method was used to detect the significant effect of the different methods on classification accuracy. If this ANOVA is significant, we would further use pair-wise t-tests to identify significant differences in recognition results between the proposed method and the other compared methods. It can be considered that there is a significant difference in the classification effect between the two algorithms when the calculated $p$-value is less than 0.05.

# RESULTS

## Experimental results

In this research, we employed the Gaussian kernel function with 5-fold cross-validation to search for optimal parameters and obtain the required experimental results. MOGWO was applied to optimize a Gaussian kernel function parameter $\lambda$ and two penalty parameters $c_1$ and $c_2$ of the classifier model. And the parameter $\lambda$ was selected from 0.001 to 8, the range of the parameters $c_1$ and $c_2$ were [0.01, 8]. The MOGWO parameters used in this paper are listed in Table 1, and the details of the parameters are given in document (*Mirjalili et al., 2016*). The Pareto optimal set obtained by MOGWO and the corresponding parameters of the proposed classifier model from subject 4 are given in Table 2. Through analyzing the Pareto optimal set, we selected the optimal solution to obtain the required final recognition accuracy. The optimal solution of subject 4 is marked in Table 2.

Next, we obtained the Pareto optimal set of the remaining subjects through the above steps and the required recognition accuracy of each class and the required average recognition accuracy we got is shown in Fig. 4. We obtained the highest classification accuracy of 98.33% from subject 7, and the average classification accuracy of all subjects is 91.27%. Besides, we calculated precision and recall to quantify the classification results. The obtained results are listed in Table 3. In order to intuitively show the experimental results of 3-class MI tasks, we drew the average confusion matrix of all subjects for the proposed method. As can be seen from Fig. 5, shoulder abduction shows the best classification accuracy (95.71%). In the meantime, through analyzing the classification accuracy of each class MI task, we find that the recognition accuracy of shoulder abduction from five subjects (except for subject 1 and subject 3) is higher than that of shoulder extension and flexion from Fig. 4. From the above results, we can consider the three classes of MI tasks

**Table 2** Pareto optimal set of subject 4 and the corresponding parameters of twin support vector machine (TWSVM).

| $c_1$ | $c_2$ | $\lambda$ | $CR_1$ | $CR_2$ | $CR_3$ | *Accuracy* |
|---|---|---|---|---|---|---|
| 2.2342 | 4.3876 | 3.8087 | 1.00 | 0.80 | 0.80 | 0.8667 |
| 2.6221 | 1.8131 | 5.2273 | 0.90 | 0.90 | 0.85 | 0.8833 |
| 2.6688 | 0.7101 | 4.7547 | 0.85 | 0.65 | 0.90 | 0.8000 |
| 2.1949 | 1.1183 | 0.1383 | 0.35 | 0.15 | 1.00 | 0.5000 |
| 2.3311 | 1.3203 | 3.5416 | 1.00 | 0.75 | 0.85 | 0.8667 |
| **2.3229** | **1.7539** | **4.3890** | **0.95** | **0.85** | **0.85** | **0.8833** |
| 2.4102 | 0.8047 | 5.4034 | 0.65 | 0.70 | 0.90 | 0.7500 |
| 2.6035 | 1.8690 | 4.6250 | 0.95 | 0.90 | 0.80 | 0.8833 |
| 2.1782 | 0.7592 | 5.2358 | 0.60 | 0.70 | 0.95 | 0.7500 |

**Notes.**
The optimal solution is marked in bold.

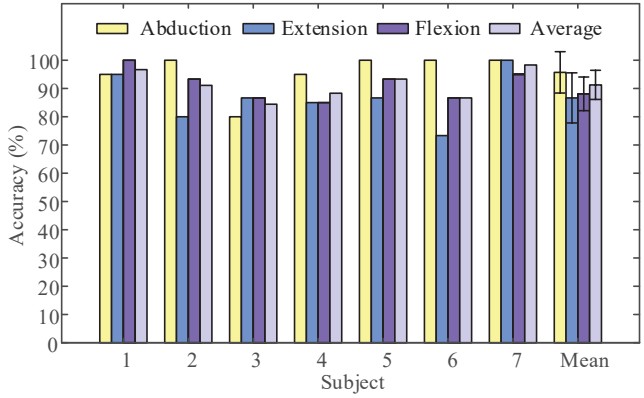

**Figure 4** **Average classification accuracies for the three categories.** Three motor imagery (MI) tasks classification accuracy of all subjects obtained using the proposed method (LMD-CSP and MOGWO-TWSVM).

can be effectively recognized by the proposed method, and shoulder abduction is easy to classify efficiently.

## Verification of feature extraction capability

To validate the extraction capability of the proposed feature extraction method, we compared LMD-CSP with some widely used feature extraction methods, including TDP (*Tavakolan et al., 2017*), CSP (*Selim et al., 2018*), FBCSP (*Ang et al., 2008*), and EMD-CSP (*Wang et al., 2008*). In our work, we used TDP, CSP, FBCSP, EMD-CSP, and LMD-CSP on our data sets for feature extraction, and the same classifier (MOGWO-TWSVM) was employed for recognition. The recognition results are shown in Fig. 6. The average classification accuracy of LMD-CSP (91.27% ± 5.16%) is higher than that of TDP (65.95% ± 13.97%), CSP (77.62% ± 4.79%), FBCSP (80.47% ± 4.97%) and EMD-CSP (84.81% ± 3.92%). Through analyzing the recognition rate of each subject, we find that LMD-CSP is higher than the compared methods among all subjects. And a one-way ANOVA revealed

**Table 3** Averaged precision and recall of three class motor imagery (MI) tasks under the proposed method.

| Subject | Precision (%) | Recall (%) |
|---|---|---|
| subject 1 | 96.75 | 96.67 |
| subject 2 | 91.55 | 91.11 |
| subject 3 | 85.15 | 84.44 |
| subject 4 | 88.65 | 88.33 |
| subject 5 | 93.45 | 93.33 |
| subject 6 | 87.05 | 86.67 |
| subject 7 | 98.41 | 98.33 |
| average | 91.57 | 91.27 |

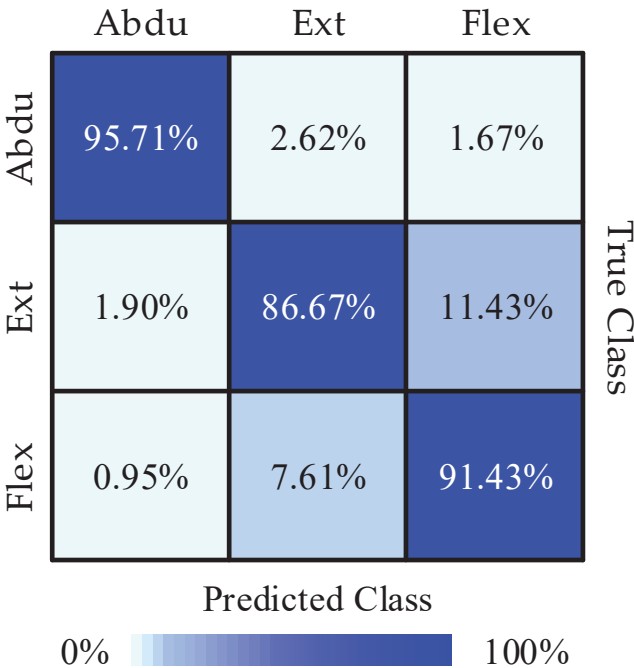

**Figure 5** The mean confusion matrix of all subjects (Abdu represents abduction, Ext represents extension, Flex represents flexion).

that there was a statistically significant difference in mean classification accuracy between at least two feature extraction methods (F (4, 30) = [10.331], $p$-value < 0.001). Pair-wise t-tests were further applied to calculate $p$-values between LMD-CSP and the other four feature extraction methods. The results are listed in Table 4. Combined the analysis of Fig. 6 and Table 4, show that LMD-CSP has stronger feature extraction capability than the other methods.

Because each MI task activated the corresponding region of the cortex, we further analyzed the contribution of different channels in the classification of the MI tasks. Since there were 3-class MI tasks, we obtained three pairs of MI tasks: abduction and extension
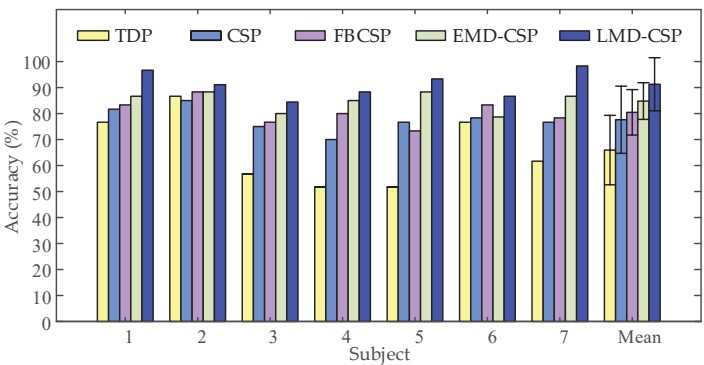

**Figure 6  Comparison of classification accuracy obtained by the different feature extraction methods.** Five different feature extraction methods, including time domain parameters (TDP), common spatial pattern (CSP), filter-bank common spatial pattern (FBCSP), common spatial pattern based on empirical mode decomposition (EMD-CSP), and the proposed feature extraction method (LMD-CSP), were employed to extract motor imagery (MI) features on our data sets, respectively. Then, the same multi-objective grey wolf optimization twin support vector machine (MOGWO-TWSVM) classified those MI features to obtain classification accuracy of all subjects.

**Table 4  Paired $t$-test ($p$-value) between the proposed feature extraction method (LMD-CSP) and the other four feature extraction methods.**

| Method | TDP | CSP | FBCSP | EMD-CSP |
|---|---|---|---|---|
| $p$-value | 0.0022 | 0.0001 | 0.0013 | 0.0141 |

(A/E), abduction and flexion (A/F), and extension and flexion (E/F). For each pair of MI tasks, the channel weight scores were calculated to highlight the maximally discriminable channels. The weight score of each channel is defined as the ratio of the 2-norm of the corresponding column vector of the CSP filter to the 2-norm of the CSP filter. More details can be seen at *Zhou et al. (2015)*. It should be noted that the weight scores were averaged across subjects and then normalized into the range (0, 1). The obtained results are shown in Fig. 7. As can be seen from Fig. 7, the prefrontal channels (AF3 and AF4), frontal channels (F3 and F4), and temporal channels (P8 and T8) contribute greatly to the classification for almost each pair of MI tasks. In *Tavakolan et al. (2017)*, the authors stated that the MI tasks of the same limb activated the adjacent cortex. Since the three MI tasks decoded in this paper belong to the same limb, these results are expected.

## Verification of classification performance

To demonstrate the performance of the proposed classifier, we input the MI feature vectors extracted by LMD-CSP into LDA, ELM, KNN, SVM, LS-SVM, and MOGWO-TWSVM for classification. The obtained results are shown in Fig. 8. It can be seen from Fig. 8 that when compared with the above conventional classifiers, the average classification accuracy by MOGWO-TWSVM is improved, increasing by 18.48%, 14.68%, 10.24%, 7.07% and 9.52%, respectively. And the difference between the average accuracy of the different classifiers was significant (F (5, 36) = [9.067], $p$-value < 0.001). The $p$-values between
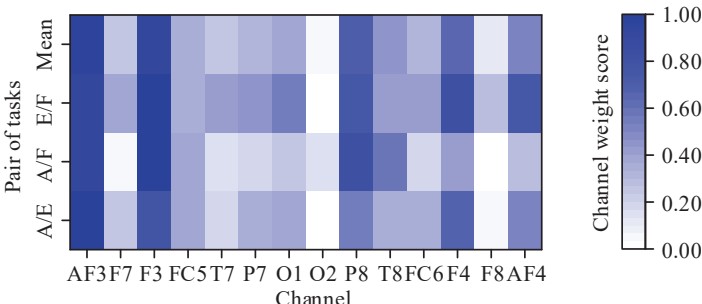

**Figure 7** Averaged channel weight scores for each pair of motor imagery tasks (A/E represents abduction and extension, A/F represents abduction and flexion, and E/F represents extension and flexion).

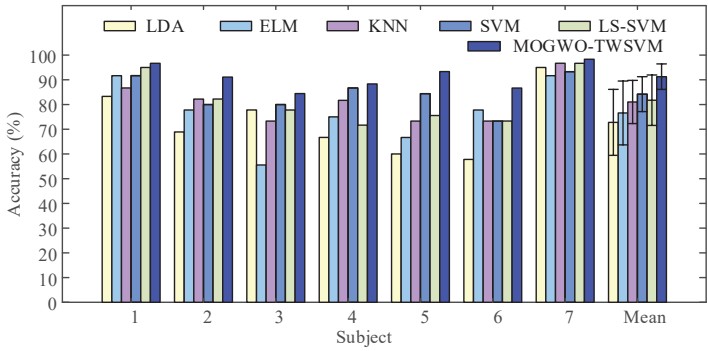

**Figure 8** Comparison of classification accuracy obtained by the different classifiers. The proposed feature extraction method (LMD-CSP) was employed to extract (motor imagery) MI features on our data sets. Then the same features were classified by six different classifier, including linear discriminant analysis (LDA), extreme learning machine (ELM), k-nearest neighbors (KNN), least squares support vector machine (LS-SVM), and the proposed classifier (MOGWO-TWSVM), to obtain classification accuracy of all subjects.

**Table 5** Paired $t$-test ($p$-value) between multi-objective grey wolf optimization twin support vector machine (MOGWO-TWSVM) and the other five classifiers.

| Method | LDA | ELM | KNN | SVM | LS-SVM |
|---|---|---|---|---|---|
| $p$-value | 0.0002 | 0.0033 | 0.0009 | 0.0005 | 0.0186 |

MOGWO-TWSVM and the other five classifiers were further calculated based on pair-wise t-tests. The results are listed in Table 5. Combined the analysis of Fig. 8 and Table 5, show that MOGWO-TWSVM has good recognition performance and strong robustness.

## Comparison with other recent methods

To demonstrate the validity of the proposed framework, we applied the proposed framework (LMD-CSP and MOGWO-TWSVM) and other recent methods such as the temporal filter parameter optimization with CSP (TFPO-CSP) (*Kumar & Sharma, 2018*) and the frequency-based deep learning scheme for recognizing brain wave signals

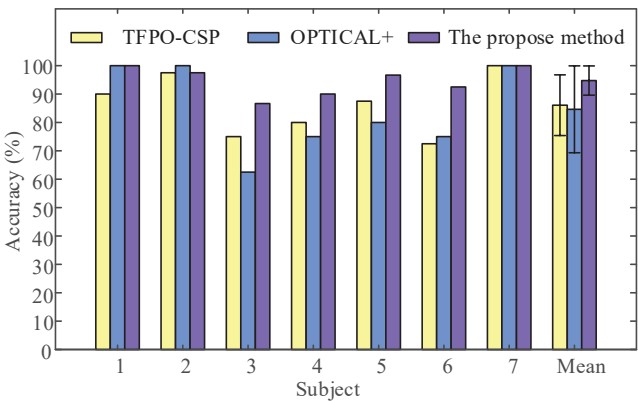

**Figure 9** Classification accuracy comparison by the proposed method with other recent methods. The temporal filter parameter optimization with CSP (TFPO-CSP), the frequency-based deep learning scheme for recognizing brain wave signals (OPTICAL+), and the proposed method were employed to discriminate two-class EEG data (shoulder abduction and extension) in our data sets to obtain the classification accuracy of all subjects.

(OPTICAL+) (*Kumar, Sharma & Sharma, 2021*) on the same data sets, this data sets include the EEG data of shoulder abduction and extension. The obtained results are shown in Fig. 9. It can be seen that the mean classification accuracy of the proposed framework (94.76% ± 5.15%) is higher than that of TFPO-CSP (86.07% ± 10.69%) and OPTICAL+ (84.64% ± 15.30%). And the average recognition rate differed significantly by framework (F (2, 18) = [3.694], *p*-value = 0.045). Pair-wise t-tests were further applied for multiple comparisons. The *p*-values between the proposed framework and TFPO-CSP and OPTICAL+ were all less than 0.05. The above results confirm the validity of the proposed method.

## DISCUSSION

In this paper, three MI tasks of the same joint are successfully recognized by using LMD-CSP and MOGWO-TWSVM with seven healthy subjects. In this section, we will discuss the proposed framework in terms of MI tasks, feature extraction, classification, limitations, and future research lines.

There are few studies on detecting the MI of different movements within the same joint. Therefore, we can only discuss studies that are similar in content to our study. Tavakolan et al. classified three different states (grasp MI, elbow MI, and rest) with an average classification accuracy of 74.2%, using time domain parameters (TDP) including autoregressive model coefficients, root mean square, and waveform length as features and an SVM classifier (*Tavakolan et al., 2017*). The weakness of that study is that it only identified different joints within the same limb. *Mammone, Ieracitano & Morabito (2020)* reported an average accuracy of 62.47% for classifying various movements (elbow flexion/extension, forearm pronation/supination, and hand open/close), using time-frequency (TF) maps as features and a deep convolutional neural network (CNN). However, multi-class MI

recognition of the same joint was not involved. In this study, we discriminated 3-class MI tasks of the same joint by using LMD-CSP for feature extraction and MOGWO-TWSVM for classification, the classification accuracy in seven subjects was no less than 84.44%. Compared with other research, we have investigated the classification of the same joint multi-class MI, and obtained higher recognition accuracy. Due to the success of discriminating single joint MI tasks, more control commands can be provided for BCI systems. And, users can intuitively control external devices, such as a robotic arm, which is of great significance for developing high-performance BCI systems.

The traditional CSP, which is widely applied to extract MI features, requires a lot of electrodes and lacks frequency information. To make up for the lack of frequency information, wavelet packet decomposition (WPD) and CSP were combined to extract effective features (*Yang et al., 2016*). That study achieved an average classification rate of 88.66% for left and right hand MI tasks. However, the wavelet basic functions need to be set manually. *Kumar & Sharma (2018)* employed the genetic algorithm (GA) to select filter parameters, which improved the performance of CSP with a classification error rate of 10.19%. *Li et al. (2016)* combined orthogonal empirical mode decomposition (OEMD) and a bank of FIR filters to enable the extracted CSP features with frequency domain information. The weakness of that study is that the proposed method has high complexity. In this study, a new adaptive time-frequency analysis algorithm, LMD, which has low computation complexity, and the traditional CSP were combined to extract MI features. LMD-CSP not only makes up for the lack of frequency information, but also improves the adaptability of the algorithm.

The purpose of this paper was to propose a framework for advanced feature extraction and classification of 3-class MI tasks within the same joint. Therefore, TWSVM was applied to classify the extracted MI features, because it has high classification speed and generalization ability. And, the parameters of TWSVM were optimized by MOGWO to improve the classification accuracy. Soman et al. classified left and right hand movements with a kappa value of 0.526, using CSP and TWSVM (*Soman & Jayadeva, 2015*). The low kappa value that can be contributed to the hyperparameter selection problem of TWSVM was not considered in that study. In another study, *Li et al. (2017)* employed CSP to extract features from left and right hand MI EEG signals, and the extracted features were input into the chaotic particle swarm optimization twin support vector machine (CSPO TWSVM), the mean accuracy was 75.95%. Compared with the approaches presented in that study, the proposed classifier (MOGWO-TWSVM) achieved a high classification accuracy (91.27%). The reason may be that MOGWO has high convergence and coverage, which can find the optimal parameters of the TWSVM to achieve the best classification performance.

It should be noted that there are some limitations to this study. On the one hand, we classified the MI tasks by using the same frequency band for different subjects in this study, which limited the classification performance of the proposed framework on some subjects. On the other hand, the sample size for each class in our data set was not large, so detailed statistical analyses were not performed.

In future research, we will verify the effectiveness of the proposed framework by using more experimental EEG data. And, we will try to decode multi-class MI tasks of distal joints,
such as the wrist joint and finger joint. Additionally, to improve classification performance, we will combine EEG and electromyography (EMG) that are related to movement.

## CONCLUSIONS

This paper proposes a new scheme for MI recognition of the same joint based on LMD-CSP and MOGWO-TWSVM. The proposed method combines LMD and CSP to extract MI features, where the cloud model is introduced to select effective PF components to enhance the separability of features, thereby successfully extracting feature vectors with high discrimination. Second, this paper employs MOGWO to tune the hyperparameters of TWSVM to improve the classification performance and generalization ability of the classifier model, and the average recognition accuracy reaches 91.27%. In future work, we will try to apply the proposed method to complex MI task recognition.

## ACKNOWLEDGEMENTS

We thank all the subjects who participated in the experiment. We thank Kai Zhao for his guidance on the EEG data acquisition.

### Funding

This work was supported by the National Natural Science Foundation of China (No.31772059), the Northeast Electric Power University (BSJXM-201521), and the Jilin City Science and Technology Bureau (20166012). The funders had no role in study design, data collection and analysis, decision to publish, or preparation of the manuscript.

### Grant Disclosures

The following grant information was disclosed by the authors:
National Natural Science Foundation of China: No. 31772059.
Northeast Electric Power University: BSJXM-201521.
Jilin City Science and Technology Bureau: 20166012.

### Competing Interests

The authors declare there are no competing interests.

### Author Contributions

- Shan Guan conceived and designed the experiments, analyzed the data, prepared figures and/or tables, and approved the final draft.
- Jixian Li conceived and designed the experiments, performed the experiments, analyzed the data, prepared figures and/or tables, authored or reviewed drafts of the paper, and approved the final draft.
- Fuwang Wang conceived and designed the experiments, performed the experiments, prepared figures and/or tables, authored or reviewed drafts of the paper, and approved the final draft.

- Zhen Yuan and Xiaogang Kang performed the experiments, authored or reviewed drafts of the paper, and approved the final draft.
- Bin Lu performed the experiments, prepared figures and/or tables, authored or reviewed drafts of the paper, and approved the final draft.

## Human Ethics

The following information was supplied relating to ethical approvals (i.e., approving body and any reference numbers):

The Scientific Research Ethics and Technology Safety committee of Northeast Electric Power University endorsed the study.

## Data Availability

Data are available at Zenodo:

Jixian Li, Shan Gaun, Fuwang Wang, Zhen Yuan, Xiaogang Kang, & Bin Lu. (2021). Motor imagery EEG data of the same joint [Data set]. Zenodo. https://doi.org/10.5281/zenodo.4699203.

## Supplemental Information

Supplemental information for this article can be found online at http://dx.doi.org/10.7717/peerj.12027#supplemental-information.

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
