# Peer review of "Discriminating three motor imagery states of the same joint for brain-computer interface"

_PeerJ, doi:10.7717/peerj.12027_

## Round 0.1 · original submission · Major Revisions

Both reviewers raise several major concerns. While you are free to do so, I understand if you decide to not apply your approach to other datasets as part of this manuscript. However, I do think that all other points should be addressed in your revision. Importantly, please include all requested information on subjects, EEG (pre)processing, feature extraction/classification, and statistical analysis.

In addition to the reviewers' comments, I think your argument for channel reduction is not convincing, as your study and the study by Mohseni et al. (2020) differ in too many aspects. Therefore, I would like to see the data re-analyzed with the full channel set. In a subsequent step, you could then consider checking which channels contribute most to classification performance in your setup. Moreover, please carefully check the manuscript for mistakes in the figures/figure labels/figure captions and make sure that the figure captions contain all the information needed to understand the figure without referring to the text. For instance, there are no error bars for mean values in Figs. 6, 8, and 9; Fig. 3 shows frequencies below 8 Hz and above 30 Hz despite reported filter settings of 8-30 Hz, and in Fig. 8 CSP is red in the legend but blue in the bar plot, several figures contain abbreviations not written in long-form in the figure caption.

Finally, throughout the manuscript, there are grammar and language errors, some of which have already been pointed out by the reviewers. I therefore strongly recommend that you undertake copyediting, either by yourselves, the aid of a colleague, or by using a professional proofreading service.

Reviewer 1 ·

Basic reporting

The English used in the article needs to be improved as there are a lot of issues with grammar and sentence structure. The subject information is missing from Table 2.

Experimental design

The experimental design looks good, with an identified gap as low recognition rate for MI of same joint. However, this needs more insight and reference. I believe that authors should evaluate other recent works using their dataset and compare them with their work.

Validity of the findings

More analysis needs to be performed to show that the proposed method is superior to existing methods of EEG signal classification. This can be done by comparing the proposed method with other recent works also evaluated using publicly available datasets.

Additional comments

The authors mention that classification of motor imagery (MI) signals relating to same joint are more difficult compared to that of MI signals belonging to different joints. LMD is proposed for signal decomposition and CSP for feature extraction, while twin support vector machine (TWSVM) is used for classification whose parameters are tuned using multi-objective grey wolf optimizer (MOGWO). An average accuracy of 89.84% is obtained using the data collected, which comprises of 7 subjects.
1. A major concern is that the paper only looks at the classification of MI signals of the same joint using a very small dataset. However, the authors should also evaluate their method on other publicly available datasets (such as BCI Competition IV dataset, EEG GigaDB dataset, etc) and compare with other related works such as those listed below, which have been published in recent years. This is imperative to show that the proposed approach not only works well on the given dataset but also performs well on other datasets for different MI tasks.
• S. Kumar, R. Sharma, A. Sharma, “OPTICAL+: a frequency-based deep learning scheme for recognizing brain wave signals,” PeerJ Computer Science, 7:e375, 2021
• M. Miao, A. Wang, and F. Liu, "A spatial-frequency-temporal optimized feature sparse representation-based classification method for motor imagery EEG pattern recognition," Medical & Biological Engineering & Computing, vol. 55, pp. 1589-1603, February 2017.
• S. Kumar and A. Sharma, "A new parameter tuning approach for enhanced motor imagery EEG signal classification," Medical & Biological Engineering & Computing, April 2018.
2. It is mentioned that the entropy (En) and super entropy (He) of the cloud model are employed to select effective PF components. However, only PF1 obtained from each channel is used for further processing. This is a one-time evaluation and not an adaptive selection method. The authors should clearly state this in the paper.
3. No source code is provided. It is highly recommended that the source code for the proposed method be made available as it will enable future research to easily test and compare the proposed method on other datasets as well.
4. Details of twin SVM should be discussed in detail.
5. Authors should also discuss other recent works published in the introduction such as those mentioned above and a few others for the benefit of the readers.
6. The exact number of spatial filters selected should be clearly stated.
7. There are a number of issues with the English used, a few are highlighted in the annotated document. A few other comments can also be found in the annotated document.

Annotated reviews are not available for download in order to protect the identity of reviewers who chose to remain anonymous.

Reviewer 2 ·

Basic reporting

- Overall, the abstract focuses too much on methods and not enough on the main finding and the significance thereof. The general framework for the proposed decoding technique is properly presented, however, detailed steps such as those involving entropy are not necessary. Furthermore, the authors should briefly mention the significance of the main result (accuracy) relative to other methods and concisely describe its potential influence in the field.

- Line 32: BCI systems "extract" sensorimotor rhythms, etc., rather than "include"

- Line 36: EEG is not random, but rather "noisy"

- Line 39 - 52: The references used to support the different feature extraction and classification methods are a bit excessive. I would consider consolidating these large lists using review articles that go in depth into these topics.

- Line 71: The LDM-CSP and MOGWO-TWSVM acronyms needs to be written in long form before being used in short form.

- Line 78: It is customary to have a "participants" section or something similar describing the human subjects in detail. At minimum, the authors should describe the sex, age and handedness of these participants. It should also be indicated if informed consent was provided and if this study was approved by an ethics board.

- Figure 1: It would be beneficial to include a visual depiction of the tasks described by the different cues.

- Figure 3-4: I am unsure of how Figures 3 and 4 contribute to the narrative of the current work. These figures provide intermediate and detailed information regarding the methods proposed, but in my opinion, do not add much to the overall results. I would be fine if these were removed or moved to supplementary information.

- Overall, the figures are aesthetically pleasing, are well organized and convey clear messages.

- In general, the English of this article is good but not great. I feel that this work could benefit from a professional proofreading service to correct various language and grammar errors.

Experimental design

- Lines 67-69: The motivation for the current work is unclear in the Introduction. They appear to motivate the current work in lines 67-69, however, as written this language is not strong enough. Why are the authors proposing this new framework? Do current techniques not produce high enough accuracies? Also, what imagery tasks are included in this work and why were they chosen?

- Line 87: please indicate if the 60 trials were the total number or per motor imagery task (i.e. 180 total).

- Line 97: The authors mention no preprocessing steps involved in EEG artifact rejection. This is particularly important as the frontal (C/FC) electrodes used in the current work are specifically prone to eye/muscle artifacts. The authors should describe if and which artifact removal techniques were employed.

- Line 147: The authors should provide justification for the use of CSP with only four EEG channels. CSP is most commonly utilized with large/full EEG montages where spatial dependencies of motor imagery tasks can be leveraged and may be less valid with so few channels. Can the authors please justify this aspect of the work with theory and or previously published works?

- Line 235: The rigor of comparing the proposed technique to numerous traditional techniques in the field is appreciated. Furthermore, these results do well to support the quality of the current work, however, I think this validation should be highlighted much earlier in the manuscript. It would be beneficial and welcomed if the authors were to state in the abstract and introduction that they validated the proposed work against various other approaches and found their technique to outperform them. Along these lines, the validation described here and presented in Figure 8 and Figure 9 appears to be well done and clearly demonstrates the improved performance of the authors' work.

- Related to my previous comment: It is very good that the authors provide statistical evidence in Table 4 to support the performance increase of their proposed approach compared to other techniques (Figure 9). However, the statistical analyses are not described until the last paragraph of the results, rather than in the methods. The authors should correct this and add a brief "Statistical Analysis" section at the end of the methods to describe how these results were obtained. Additionally, it sounds like the authors applied a one-way ANOVA two each pairwise combination of their proposed method and one other traditional method to evaluated significance. If this is the case, I feel that this is an inappropriate statistical test. The authors should in fact apply a one-way ANOVA to all six methods and then post-hoc pair-wise t-tests/Tukey HSD tests. In this case, the authors only need to report the results of the pair-wise tests involving their proposed method. While I dont expect the overall results to change much at all, I feel that this would be a more appropriate methodology.

Furthermore, is there a reason why the authors do not provide the same statistical results for the the values presented in Figure 8? Also, is it necessary to include a table(s) with statistical results when they can also be added to the group-level bar graphs in Figures 8 and 9?

- The main results in the current work involve classification accuracies or similar/related metrics. Would it be possible for the authors to include and physiological results such as the CSP maps that indicate the weighted importance of the different electrodes used?

Validity of the findings

- In the Discussion, the authors need to describe how this work compares to similar work published in the field. Obviously, the authors will highlight the within data set validation that they performed themselves, but will also need to describe the classification performance of similar tasks investigated by other groups/studies. For example, how do these results and tasks used compare to the rest of the field? If there are large discrepancies, potential reasons should be provided.

As is it currently written, the Discussion appears to be repetitive with the methods section and dedicates too much content to the technical details. The Discussion should be of high-level than this and focus on where the results fall within the field at large and how they may impact it in the future. I would therefore suggest that the very large second paragraph of the Discussion to be reworked to accommodate these suggestions.

---

## Round 0.2 · Minor Revisions

Both reviewers still have minor concerns. Please address the concerns raised by Reviewer 2 in your resubmission. Regarding the new Figure 7, in addition to the comments by Reviewer 2, I suggest changing the way in which channel weight scores are visualized. The interpolation used to create the topographic plot from a small and unevenly distributed set of channels creates the false impression that there are data for scalp regions where in fact there are non. The concern raised by Reviewer 1 should also be addressed in your resubmission, or at least be commented on in your rebuttal letter.

Reviewer 1 ·

Basic reporting

Much improved compared to previous submission

Experimental design

Acceptable

Validity of the findings

Performance of the proposed systems as reported looks good.

Additional comments

The comparison made in the revised version (lines 282-294) is not valid. One cannot compare different methods by evaluating them using different datasets. The authors should consider revising the comparisons made.

Reviewer 2 ·

Basic reporting

Overall, the work has been markedly improved during this recent round of revisions. I believe all of my previous comments from this section have been properly addressed.

Experimental design

- I agree with the editor that the new analysis with the full channel montage is far more valid than the original analysis using only four channels. Furthermore, this step seemed to improve the results overall. Due to the change, however, I feel that the following sentence should be removed from the introduction as channel number is no longer a significant advantage of the current work compared to others.

"However, it requires a larger number of input channels in that study."

- I do not feel that my previous concern regarding the statistical analysis was properly addressed. In the original submission the authors performed pairwise ANOVAs between the classification accuracies of different methods. By contrast, in the current revision, the pair-wise ANOVAs were simply changed to pair-wise t-tests. I believe that this approach could be valid if the authors correct the individual t-test values for multiple comparisons (i.e. Bonferroni or false discovery rate). Alternatively, as I mentioned in the previous round of review, I feel that an ANOVA containing the classification accuracies of all methods is necessary before the pair-wise t-tests. As long as a significant difference exists among any of the included methods, this ANOVA should be significant and would warrant subsequent pairwise t-tests. I recommend that the authors choose to perform one of these options in the following round of review.

- Please add units to the color bar of the new Figure 7. There should also be more details regarding how this group-level topography was created. In the authors' response to reviewer 1 Q6, they state that three spatial filters were obtained for classification (one for each task pair). Ideally, I assume that each one of these filters contains weights to maximize the between-class variance of any two MI tasks. Therefore, it seems more reasonable to show average channel weights for each MI task pair (x 3) to highlight the channels that are maximally discriminable between the different tasks rather than simply their weighted "importance". I think this would provide a deeper level of understanding regarding the influence of different channels on the neural representation of the various MI tasks.

- I feel that further English edit is still necessary throughout the manuscript and that a professional proofreading service would be beneficial.

Validity of the findings

I believe all of my previous comments from this section have been properly addressed.

---

## Round 0.3 · Minor Revisions

The manuscript has considerably improved again following the second round of reviews.

In your rebuttal letter, you ask for guidance regarding the one-way ANOVAs that you conduct. From what I see in your manuscript, you now follow the suggestion by reviewer 2 and run ANOVAs before the post-hoc t-Tests. However, you do not yet report the results of the ANOVAs, which is a necessity. Guidelines on how this reporting should be done can be found online (for instance here: https://www.statology.org/how-to-report-anova-results/), another very useful resource that I can recommend is the Publication Manual of the American Psychological Association. Please take these guidelines into consideration and report your statistical results accordingly in your resubmission.

Also, please avoid using the word 'significantly' ("... temporal channels (P8 and T8) contributed significantly to the ...") when not describing statistical results to avoid confusion.

---

## Round 0.4 · accepted · Accept

I have no further comments.